# Molecular and Crystal Structure of a Chitosan−Zinc Chloride Complex

**DOI:** 10.3390/nano11061407

**Published:** 2021-05-26

**Authors:** Toshifumi Yui, Takuya Uto, Kozo Ogawa

**Affiliations:** 1Faculty of Engineering, University of Miyazaki, Nishi 1-1 Gakuen-kibanadai, Miyazaki 889-2192, Japan; 2Organization for Promotion of Tenure Track, University of Miyazaki, Nishi 1-1 Gakuen-kibanadai, Miyazaki 889-2192, Japan; t.uto@cc.miyazaki-u.ac.jp; 3Research Institute for Advanced Science and Technology, Osaka Prefecture University, 1-2 Gakuencho, Sakai, Osaka 599-8570, Japan; ogawakt@kawachi.zaq.ne.jp

**Keywords:** chitosan−ZnCl_2_ complex, crystal structure, X-ray fiber diffraction

## Abstract

We determined the molecular and packing structure of a chitosan–ZnCl_2_ complex by X-ray diffraction and linked-atom least-squares. Eight D-glucosamine residues—composed of four chitosan chains with two-fold helical symmetry, and four ZnCl_2_ molecules—were packed in a rectangular unit cell with dimensions *a* = 1.1677 nm, *b* = 1.7991 nm, and *c* = 1.0307 nm (where *c* is the fiber axis). We performed exhaustive structure searches by examining all of the possible chain packing modes. We also comprehensively searched the positions and spatial orientations of the ZnCl_2_ molecules. Chitosan chains of antiparallel polarity formed zigzag-shaped chain sheets, where N2···O6, N2···N2, and O6···O6 intermolecular hydrogen bonds connected the neighboring chains. We further refined the packing positions of the ZnCl_2_ molecules by theoretical calculations of the crystal models, which suggested a possible coordination scheme of Zn(II) with an O6 atom.

## 1. Introduction

Chitin—a linear polysaccharide composed of β-(1→4)-linked *N*-acetyl-D-glucosamine monomers—is the most abundant and renewable natural polymer after cellulose. One commonly finds chitin in the exoskeleton or cuticles of many invertebrates, and the cell walls of most fungi and some algae. Recently, chitinous skeletal fibers from some marine demosponges were attracted attention as a possible application to scaffolds for cultivation [1]. Chitosan, a partially or fully deacetylated derivative of chitin, exhibits a regular distribution of aliphatic primary amino groups and primary hydroxyl groups. Unlike its corresponding parent polymer, chitosan is soluble in various aqueous acids and has a remarkable ability to form specific complexes with a number of ions, such as transition and post-transition metal ions [2,3]. Chitosan is a unique cationic biopolymer that is available in large quantities. Researchers have commercialized it as follows: a coagulant for wastewater, animal feed, fertilizer, an antibacterial additive for clothing, and a precursor for glucosamine [4]. Chitosan is conventionally a powder or a film, yet chitosan nanofibers—typically fabricated by electrospinning—are an active area of research. Researchers have used electrospun nanofibers of chitosan—and corresponding blends with other polymers—for separations, as biological scaffolds, and for drug delivery [5,6,7]. Researchers have also used electrospun chitosan nanofibers to remove low concentrations of metal ions [8] and arsenate ions [9] from water.

Researchers first obtained the X-ray fiber pattern of chitosan—by solid-state deacetylation of a lobster tendon—in a hydrated crystalline form [10]. The results indicated a two-fold helical symmetry for chitosan as a molecular chain structure in a similar manner as chitin. Researchers first obtained the other polymorph of chitosan, an anhydrous form, by annealing the following: a stretched chitosan sample at ≥190 °C in water [11], and a crab tendon chitosan in water at ~240 °C [12]. The reported crystal structures of both the hydrated and anhydrous forms are in atomistic detail, as per analyses of corresponding X-ray diffraction data [12,13,14]. Researchers reported further structural details of the anhydrous form—such as the positions of the hydroxyl and amino hydrogen atoms—by using periodic density functional theory (DFT), which indicated a possible scheme of the hydrogen bond network [15]. The structures of chitosan metal complexes and salts are also pertinent. Researchers have proposed two types of coordination mode: the pendant model [16,17] and the bridge model [18]. The chitosan–HI salt, the only chitosan complex for which there is a crystal structure in atomistic detail, involves two independent iodide ions: one coordinated with three amino groups, and one that accepts one hydrogen atom from an amino group and two hydrogen atoms from the O6 hydroxyl groups [19]. This may be interpreted as a hybrid of the bridge and pendant coordination modes, in terms of interactions with the amino groups. However, the coordination structures of chitosan–metal complexes remained unsolved.

Ogawa et al. reported an X-ray diffraction study of crab tendon chitosan samples complexed with Cd(II), Zn(II), and Cu(II) salts; the ZnCl_2_ complex sample provided a fiber diffraction pattern of higher quality compared with the other chitosan metal salts [20]. The affinity of chitosan for Zn cations can be pertinent to applications. For example, regarding wastewater treatment, chitosan exhibits a medium affinity for Zn(II) compared with other divalent metallic ions—approximately one-third that for Cu(II) and almost equivalent to that for Cd(II), which enabled a chitosan film to remove zinc ions up to ~50% of the initial concentration in the effluent [21]. The chitosan–Zn complex exhibited broad-spectrum antimicrobial activities, especially against *Escherichia coli* [22]. Researchers incorporated a zinc electrodeposit—applied on a steel construction surface for protection and as a barrier against corrosion—with chitosan to form chitosan–zinc composite electrodeposits with enhanced antibacterial properties [23]. Nanoscale fabrication of chitosan further enhances these functionalities of zinc complexes.

In the present study, we determined and analyzed the crystal structure of a chitosan–ZnCl_2_ complex to reveal the coordination structure of the zinc ion by X-ray fiber diffraction, which we combined with molecular mechanics (MM) and quantum mechanical modeling. The final crystal structure that the molecular chains arranged to form zigzag-shaped chain sheets along the *a*-axis, where the neighboring chains were connected by intermolecular hydrogen bonds involving the N2 and O6 atoms. ZnCl_2_ molecules were located at the bending positions of the chain sheets. Although no clear coordinate bond with an amino group was detected, O6–H···Zn(II) coordinate bonds were suggested in the semi-empirical quantum mechanics (SEQM)-optimized crystal model.

## 2. Materials and Methods

### 2.1. Sample Preparation and X-ray Diffraction

Our method of obtaining X-ray fiber diffraction data was described in our previous study [16,20]. Briefly, tendon chitosan was prepared from a highly oriented chitin specimen of a crab tendon, *Chionecetes opilio* O. Fabricius, by *N*-deacetylation with 67% aqueous sodium hydroxide at 100 °C for 2 h under a nitrogen atmosphere. The degree of *N*-acetylation of the tendon chitosan was found to be 0% by measurement of a colloidal titration, and the viscosity average polymerization was 10,800 [16,20]. The tendon chitosan was soaked in aqueous ZnCl_2_. The X-ray fiber diffraction patterns were recorded with a box camera equipped with an imaging plate (Fujifilm HR-III), at 76% relative humidity in a helium atmosphere, with a Rigaku Geigerflex diffractometer equipped with Ni-filtered Cu Kα radiation. The X-ray diffraction image was read with an imaging plate detector (Rigaku R-AXIS), and the three-dimensional intensity profile was analyzed with Surfer (Golden Software, Inc., Golden, CO, USA) for resolution of overlapped profiles, background removal, and calculations of peak intensities. A set of the measured intensities was corrected for the Lorentz and polarization factors to provide a set of observed structure factors, *F*_o_.

The density of the tendon chitosan–ZnCl_2_ complex was measured by flotation with a carbon tetrachloride–*m*-xylene solution.

### 2.2. Crystal Structure Analysis

Figure 1 shows the principal parameters to describe chitosan chain conformation and position in the unit cell, together with atom labeling. A molecular chain structure with two-fold helical symmetry was the observed form of the chitosan crystal structures and was not substantially changed throughout structure refinement except in terms of the orientations of the hydroxymethyl groups, χ_O6_, which were defined by the O5–C5–C6–O6 sequence. The conformation usually prefers three staggered positions, which was termed as either *gauche*–*trans* (*gt*), *gauche*–*gauche* (*gg*), or *trans*–*gauche* (*tg*); the χ_O6_ values of the respective positions are 60°, −60°, and 180°. The positions of the ZnCl_2_ molecules were associated with the primary amino groups of the glucosamine residues and were defined by a set of internal coordinates: the N2–Zn distance, *d*_N2–Zn_; the C2–N2–Zn angle, θ_Zn_; the rotation of the C1–N2 bond, χ_Zn_; the N1–Zn–Cl1 angle, θ_Cl1_; and the rotation of N2–Zn, χ_Cl1_. The torsion angles, χ_Zn_ and χ_Cl1_, were defined by the atom sequences of C1–C2–N2–Zn and C2–N2–Zn–Cl1, respectively. The initial position of the other chlorine atom, Cl2, was defined by reporting 180° as the Cl1–Zn–Cl2 angle, such that the ZnCl_2_ molecule of an initially linear structure could bend in either direction during structure refinement.

The chain positions in the crystal unit cell were defined by the chain packing parameters: the chain rotational position, μ in degrees, with respect to the helix axis; and the chain translational positions, *u, v,* and *w*, in fractions, along the *a*, *b*, and *c* dimensions, respectively. Whereas the *u* and *v* parameters define the position of the helix axis on the *ab* base plane, the *μ* and *w* parameters correspond to the positions of the root atom, O4_root_, in Figure 1; µ = 0° and *w* = 0 when O4_root_ is at (x, 0, 0) for a helix origin.

Molecular and packing structures—of chitosan chains and ZnCl_2_ molecules—were determined by using the linked-atom least-squares (LALS) program [24,25], where the quantity Ω was minimized as per the following:(1)$$\Omega =\overset{}{\underset{m}{\sum }}w_{m}\left({\left|F_{m,o}\right|}^{2}-k^{2}{\left|F_{m,c}\right|}^{2}\right)+s\overset{}{\underset{i,j}{\sum }}\epsilon_{i,j}+\overset{}{\underset{q}{\sum }}\lambda_{q}G_{q}$$

The first summation term ensures optimum agreement between the observed (*F_m,o_*) and calculated (*F_m,c_*) X-ray structure amplitudes of the *m*-th reflection. Term *k* is a scaling factor. The weight of the reflection, *w_m_*, was fixed to 1.0 for observed reflections, 0.5 for unobserved reflections in which *F_m,c_* > *F_m,o_*, and 0.0 for unobserved reflections in which *F_m,c_* < *F_m,o_*. The second summation term evaluates non-bonded repulsions between atoms *i* and *j*. The quantity *s* is an overall weight of the non-bonded repulsions. The third summation term imposes the atomic coordinate constraints by the method of Lagrange undermined multipliers. The constraints were adopted to preserve helix continuity and pyranose ring closure of the residue. The overall agreement between the observed and calculated X-ray structure amplitudes was evaluated by a weighed residual:(2)$$R_{w}=\frac{\sum_{m}^{}w_{m}{\left(\left|F_{m,o}\right|-\left|F_{m,c}\right|\right)}^{2}}{\sum_{m}^{}w_{m}F_{o}^{2}}$$

The unobserved reflections below the observable threshold were included in calculating both Equations (1) and (2). One-half of the minimum intensity was assigned to estimate the magnitude of *F_o_* for unobserved reflections.

### 2.3. Theoretical Calculations of Crystal Models

A theoretical study of crystal models with finite dimensions was carried out to refine orientations of the hydroxyl, hydroxymethyl, and amino groups and the positions of the ZnCl_2_ molecules. Two crystal models, differing in the constituent number of the chitosan chains, were constructed on the basis of the crystal structure determined by X-ray analysis: chitohexaose × 25 and chitohexaose × 9. Both models considered four ZnCl_2_ molecules in the inner core, which enabled the ZnCl_2_ molecules to exist in a crystalline state. Partial structure optimizations were applied to the crystal models, such that the three substituent group orientations and the ZnCl_2_ positions were allowed to vary, whereas the backbone structures of chitohexaose were static. MM was applied to optimizations of the chitohexaose × 25 models by using a universal force field (UFF) [26] where the atomic charges were assigned by charge equilibration [27]. Researchers have applied a UFF to MM calculations involving metal complexes [27,28,29]. Optimizations of the chitohexaose × 9 models were performed by using SEQM with a PM6 Hamiltonian [30]. Tight convergence self-consistent field criteria (10^−8^ a.u.) were applied in the SEQM optimizations. The accuracy of the PM6 Hamiltonian for Zn(II) complexes was demonstrated by systematic benchmarks [30,31]. Its reliability was also reported for crystal structure studies of proteins and other organic materials with complexes of Zn(II) [32,33,34]. All of the MM and SEQM calculations were performed by using Gaussian 09 Rev. C01 [35].

### 2.4. Visualizations of Crystal Structures

Molecular graphics software, PyMOL 1.7.1, was used to visualize and draw the crystal structures [36].

## 3. Results and Discussion

### 3.1. Crystal Data

Figure 2 shows an X-ray fiber diffraction pattern of the chitosan–ZnCl_2_ complex. We indexed a total of 33 observed diffraction spots up to the fourth layer with a rectangular unit cell, dimensions *a* = 1.1677 nm, *b* = 1.7991 nm, and *c* =1.0307 nm (where *c* is the fiber axis). Appendix A shows the observed and calculated *d*-spacings. The unit cell accommodated eight glucosamine units—comprised of four chitosan chains with a two-fold helical conformation, and four ZnCl_2_ molecules—resulting in a calculated density ρ_calc_ = 1.41 g/cm^3^. One can insert an additional 16 water molecules into the unit cell, resulting in ρ_calc_ = 1.63 g/cm^3^, for a better fit to the observed density, ρ_obs_ = 1.56 g/cm^3^.

### 3.2. Search for Chain Packing Structures

Researchers have proposed that chitosan chains are packed with *P*2_1_2_1_2_1_ symmetry in the parent crystal structures of both the anhydrous chitosan and chitosan hydrate crystal structures, where the positioning of a pair of neighboring chains in antiparallel polarity is related by two-fold helical symmetry along either the *a* or *b* axis [12,13,14]. Similarly, we assumed involvement of two-fold helical symmetry to generate antiparallel chain pairs, despite the fact that we observed odd reflections on *h*00 and 0*k*0, such as 100 and 070, in the fiber diagram. We rationalized such violation of systematic absences by non-symmetrical packing of ZnCl_2_ molecules. We assumed two types of space groups, *P*2_1_/*a* and *P*2_1_/*b*, to define a chain packing structure in the unit cell. The space group models required two independent chains, each consisting of two consecutive glucosamine units as an asymmetric unit. We defined the eight chain packing models by a combination of the space group, chain positions, and chain polarities for the two independent chains (Table 1). For each of the chain packing models, we considered the three staggered positions of the hydroxymethyl group (*gt*, *gg*, and *tg*) to give 24 initial structures. For all eight glucosamine residues, we linked a ZnCl_2_ molecule to an amino group with an initial *d*_N2–Zn_ of 0.22 nm, whereas we set the occupancy of Zn and Cl atoms to be 0.5, and thus the number of ZnCl_2_ molecules was effectively four, instead of eight.

The first stage of the chain packing structure search was to determine the appropriate *ab* projection structures based on the *F*(*hk*0) data for the initial 24 structures. The values of μ1 and μ2 were stepped from a value of 0° to a value of 180° by 10° increments to provide two-dimensional *R*_w_ maps. We generally found the *R*_w_ minima at approximately (μ1, μ2) = (0°, 0°), (0°, 90°), (90°, 0°), and (90°, 90°) in most of the 24 *R*_w_ maps. On the basis of the results of the μ1–μ2 search, we generated 432 initial structures by combining six μ1 and μ2 values (–10°, 0°, 10°, 80°, 90°, and 100°), three χ_O6_ values (60°, –60°, and 180°), two χ_Zn_ values (–60° and 120°), and two χ_Cl1_ values (0° and 90°); and refined the structures with respect to the μ1 and μ2 values. Among the 432 μ1–μ2 refined structures, we selected that which exhibited the lowest *R*_w_ value to represent each of the chain packing models (Appendix A). As a result, chain packing model 2 clearly differed from the other models by its comparatively low *R*_w_ values. Table 2 shows the final *R*_w_ values of the three models of chain packing model 2, all of which were adopted to search for an appropriate N2–Zn distance, *d*_N2–Zn_. Because LALS sets bond length as a non-variable parameter, we carried out the *d*_N2–Zn_ search by stepping the *d*_N2–Zn_ value from a value of 0.22 nm to a value of 5.0 nm in 0.01-nm increments, whereas we varied the parameters μ1, μ2, χ_O6_, χ_Zn_, χ_Cl1_, and θ_Zn_. Table 2 shows the *R*_w_ values at the minima with respect to the *d*_N2–Zn_ change, and the corresponding *d*_N2–Zn_ values.

We then focused our crystal structure analysis on three-dimensional chain packing structure searches using the higher-layers *F*(*hkl*) data, where *l* = 1 − 3, in addition to the *F*(*hk*0) data. For each of the 10 refined models (Table 2), we stepped the chain translational positions (*w*1 and *w*2) from a value of –0.4 to a value of 0.5 in increments of 0.1, which generated 100 structures with respect to the *w*1–*w*2 positions. We varied the *w*1 and *w*2 values, and the angle parameters θ_Zn_, in addition to the parameters in the previous *d*_N2–Zn_ search. We selected the three-dimensional chain packing structure with the lowest *R*_w_ value from the 100 refined structures. Among the remaining 10 models, we screened three (*d*_N2–Zn_ = 0.30, 0.37, or 0.46 nm) for the next crystal structure refinement stage. Table 3 shows the final *R*_w_ values and the refined chain packing parameters of the three selected models. Although the initial hydroxymethyl conformations were *gt* for models 1 and 2, and *tg* for model 3 (Table 3), most of the χ_O6_ angles substantially rotated into different conformers compared with the original conformations.

### 3.3. Crystal Structure Refinement by Combined X-ray Data and Stereochemical Constraints

We performed the final stage of crystal structure analysis mainly to refine the ZnCl_2_ packing positions. We introduced the stereochemical constraints, the second term of Equation (1), to suppress development of unrealistically short contacts between nonbonding atoms during structure refinement. We excluded a hydroxymethyl conformation, χ_O6_, from two-fold helical symmetry operation of the chain, which allowed four χ_O6_ parameters—corresponding to the two independent chains—to rotate independently. In the structure refinement stage, we set the occupancy of a ZnCl_2_ molecule to be 1.0, such that the two amino groups were linked to the two respective ZnCl_2_ molecules and the remaining two remained unlinked, which required six linking patterns of ZnCl_2_ molecules (Appendix A). We allowed two sets of the ZnCl_2_ position parameters (χ_Zn_, χ_Cl1_, and θ_Zn_) to independently vary, and we stepped the remaining ZnCl_2_ position parameter (*d*_N2_**_–_**_Zn_) from a value of 0.27 nm to a value of 0.47 nm in 0.01 nm increments. At each *d*_N–Zn_ position, we generated 1296 initial structures by combining four sets of three χ_O6_ values (60°, −60°, and 180°), two sets of two χ_Zn_ values (−60° and 120°), and two sets of two χ_Cl1_ values (0° and 90°); whereas we used the initial chain packing positions (μ1, μ2, *w*1, and *w*2) from either of the three selected models (Table 3), depending on the *d*_N2–Zn_ value. We carried out full minimization of the Ω function of Equation (1) with respect to all of the parameters previously discussed. Figure 3 shows the adiabatic *R*_w_ changes for the six ZnCl_2_ linking pattern models with respect to the *d*_N2–Zn_ value, where we used the *R*_w_ values from the refined structures corresponding to the lowest *R*_w_ values among the 1296 possible values. The *R*_w_ values decreased in accordance with increasing *d*_N–Zn_, accompanied by several minima, suggesting that the ZnCl_2_ molecule preferentially intercalated between the chitosan chains rather than closely coordinated to a particular primary amino group. Table 4 shows the chain packing parameters (μ and *w*), *R*_w_, and contact σ, the second summation term of Equation (1) representing a magnitude of short nonbonding contact, for the final seven structures, which we selected on the basis of the threshold *R*_w_ value of 0.245 (defined arbitrarily). Considering the fact that there were negligible differences in the *R*_w_ values among the seven final structures, we selected model 4 because it corresponded to the smallest contact σ value corresponding to a comparatively low *R*_w_ value.

We performed an additional full optimization run for model 4, where we introduced the angle parameter, θ_Cl1_, to allow ZnCl_2_ molecules to form a bent structure; we increased the magnitude of an overall weight of the non-bonded repulsions, s, in Equation (1) by 5× that which we adopted in the preceding runs to remove excessive non-bonding repulsions. Table 5 shows the values of the representative parameters of the final structure, and Figure 4 shows the structure’s projections. Appendix A shows the coordinates of the final structure, and Appendix A shows the observed and calculated structure factor amplitudes.

Although the final *R*_w_ value increased slightly at the expense of a decreasing contact σ value, the short contact between an N2a atom in chain 3 and a Cl atom of the ZnCl_2_–3 molecule remained, resulting in a distance of 0.22 nm. The N2 and O6 atoms participated in intermolecular hydrogen bonding to form chain sheets that exhibited a zigzag shape along the *a*-axis. The distances of O6a···O6b intermolecular hydrogen bonds, ~0.2 nm, involved the chain 1–2 sheets, possibly indicating the presence of short contacts in the atom pairs rather than hydrogen bonds. The other intermolecular hydrogen bonds in the chain 3–4 sheets formed with an optimum distance of ~0.28 nm. ZnCl_2_ molecules were located around the bending portions of the chain sheets. The molecules were simply intercalated between the chain sheets and did not clearly coordinate to any particular amino or hydroxyl group nearby.

### 3.4. Theoretical Calculations of Crystal Models

We examined the crystal structure by using theoretical calculations implemented by MM and in accordance with SEQM methods, to refine the positions of the ZnCl_2_ molecules as a primary objective. We performed the first MM calculations to search for preferred orientations of the hydroxyl and amino groups. Combining three staggered positions of the two hydroxyl groups on the C2 and C6 atoms, and those of the amino group, for each of the eight independent glucosamine units generated 6561 initial structures of the 25×hexaose model. We initially set the C3 hydroxyl group to be C2–C3–O3–H at *trans*, which formed an O3–H···O5 intramolecular hydrogen bond commonly observed in the parent chitosan crystal structures [12,13]. We partially optimized the structures of the 25×hexaose models, where the backbone structures of the chitosan chains were static, whereas we allowed the following to change: the orientations of the hydroxyl, amino, and hydroxymethyl groups; and the positions of the ZnCl_2_ molecules. We selected 100 structures in terms of the lowest total steric energy and transferred their structural features (the substituent orientations and ZnCl_2_ positions) to the 9 × hexaose models, which we then subjected to SEQM partial structure optimization. Figure 5 shows the *ab* projection of the lowest structure obtained by SEQM calculations. Compared with the starting crystal structure, the molecular axes of the ZnCl_2_ molecules were more aligned with the fiber axis, whereas the Zn atoms had not moved appreciably from their initial positions. They had slightly bent to give Cl1–Zn–Cl2 angles of 147° and 152°, respectively, and the corresponding Zn atoms were coordinated by the adjacent O6–H groups, with N2–O6 distances of 0.22 and 0.24 nm, respectively. The DFT study of the D-glucosamine–ZnCl_2_ complex models predicted the length of the coordinate bond between a glucosamine residue and a ZnCl_2_ molecule to be ~0.20 nm [37]. The hydrogen-bonding network suggested for the crystal structure was found mostly conserved in the SEQM-optimized structure, where the amino groups served as hydrogen donors to the O6–H groups. Unfortunately, we observed all of the 100 SEQM-optimized structures—as per structure amplitudes—to give unacceptable *R*_w_ values of ~0.5. A possible interpretation for the considerable disagreement between the X-ray data and SEQM-optimized structure is that ZnCl_2_ molecules, in the presence of water molecules, may partly dissociate in the chitosan–ZnCl_2_ crystal structure. The crystal structures derived from concentrated ZnCl_2_ solutions depend on complete coordination of Cl^−^ and H_2_O around Zn^2+^ [38]. In the crystal structure of ZnCl_2_·2.5H_2_O, for example, one Zn^2+^ involved a tetrahedral coordination with Cl^−^, and the other Zn^2+^ resided in the octahedral environment defined by five H_2_O molecules and one Cl^−^ shared with [ZnCl_4_]^2−^ [38].

## 4. Conclusions

We reported atomistic detail of a chitosan–ZnCl_2_ crystal structure. The molecular chains arrange to form zigzag-shaped chain sheets along the *a*-axis, where the neighboring chains in antiparallel polarity—related by two-fold helical symmetry along the same axis—are connected by intermolecular hydrogen bonds involving the N2 and O6 atoms. ZnCl_2_ molecules are located at the bending positions of the chain sheets. The features are substantially similar to those detected in the crystal structure of chitosan–HI salt [19], suggesting that the features are likely common among chitosan–metal and chitosan–salt complexes. Although we did not detect a clear coordinate bond in the present crystal structure, a minor adjustment of the hydroxymethyl substituent structure may correspond to O6–H···Zn(II) coordinate bonds, in accordance with the structural features of the SEQM-optimized crystal model. It is possible that there is more stable complexation in the crystal structure involving dissociated Zn^2+^ and Cl^−^ ions, as well as H_2_O molecules, resulting in Zn(II) that is readily accessible to the amino and hydroxyl groups. Two-fold helical symmetry—introduced to the present structure analysis because of technical reasons—restricted the diversity of the molecular packing, likely preventing glucosamine residues and ZnCl_2_ molecules from complex formation. Extension to a complete *P*1 symmetry analysis may require one to investigate a large number of structural parameters and X-ray diffraction data obtained at higher resolution.

## Figures and Tables

**Figure 1 nanomaterials-11-01407-f001:**
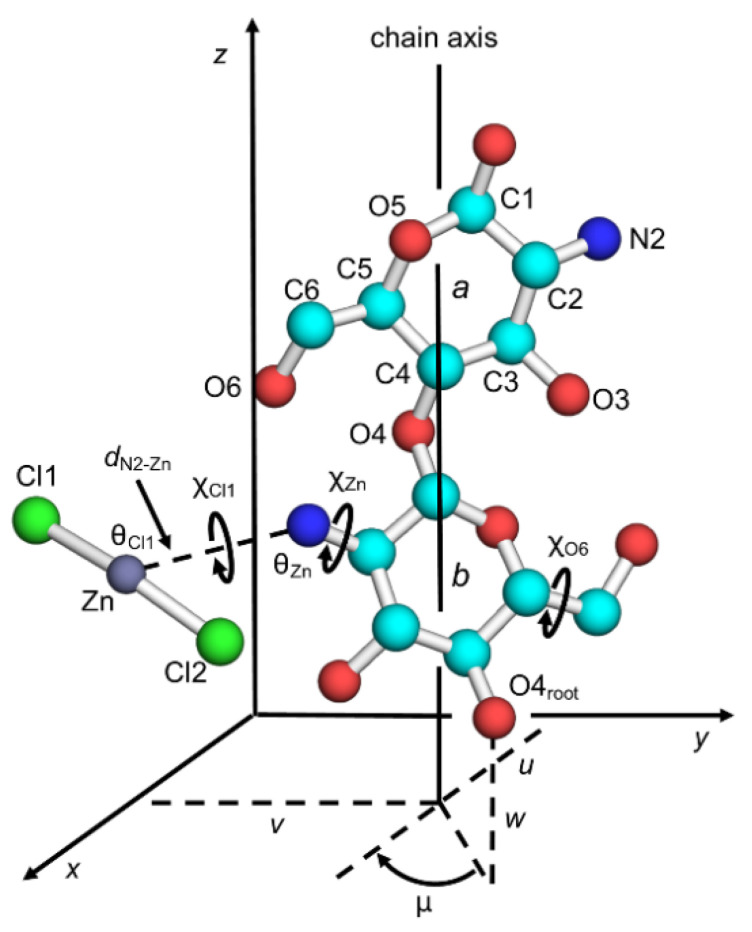
Two-fold helix structure of chitosan together with atom and residue designations. Notation is as follows: chain packing parameters (μ, *u*, *v*, and *w*), hydroxymethyl conformation (χ_O6_), and ZnCl_2_ position parameters (χ_Zn_, χ_Cl1_, θ_Zn_, θ_Cl1_, and *d*_N2–Zn_).

**Figure 2 nanomaterials-11-01407-f002:**
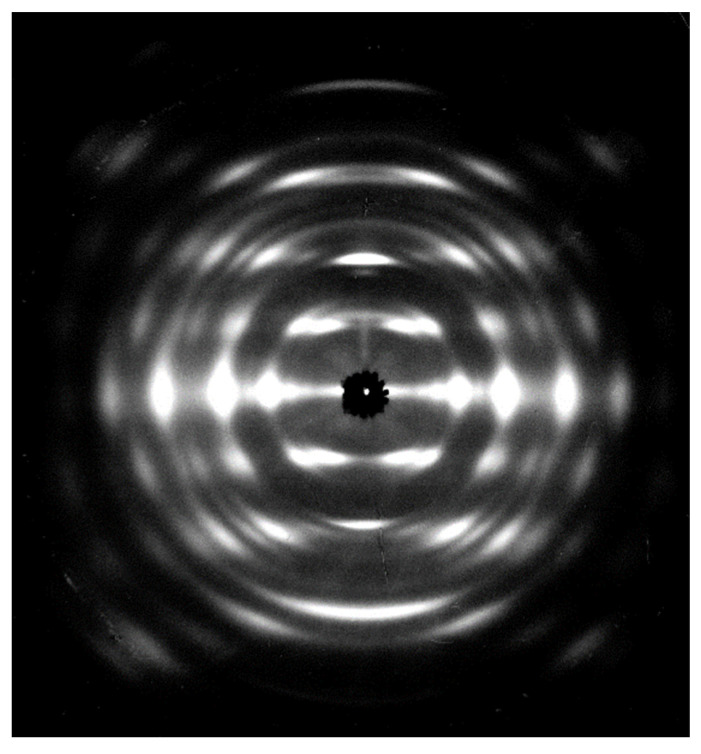
X-ray fiber diffraction pattern of the chitosan–ZnCl_2_ complex.

**Figure 3 nanomaterials-11-01407-f003:**
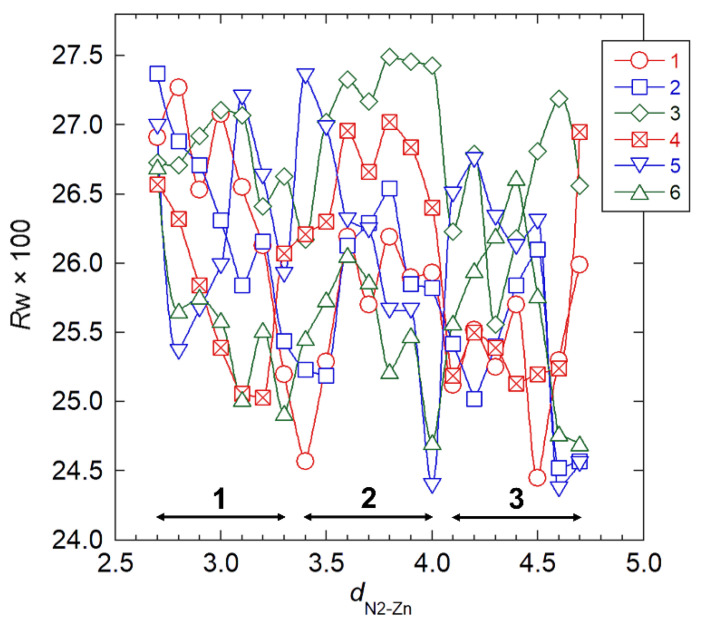
*R*_w_ with the respect to N2–Zn distance, *d*_N2–Zn_, in structure refinement for six ZnCl_2_ linking patterns. Double-headed arrows indicate *d*_N2–Zn_ ranges where we obtained the models by the preceding three-dimensional structure refinement. Numbers 1–6 refer to corresponding model numbers.

**Figure 4 nanomaterials-11-01407-f004:**
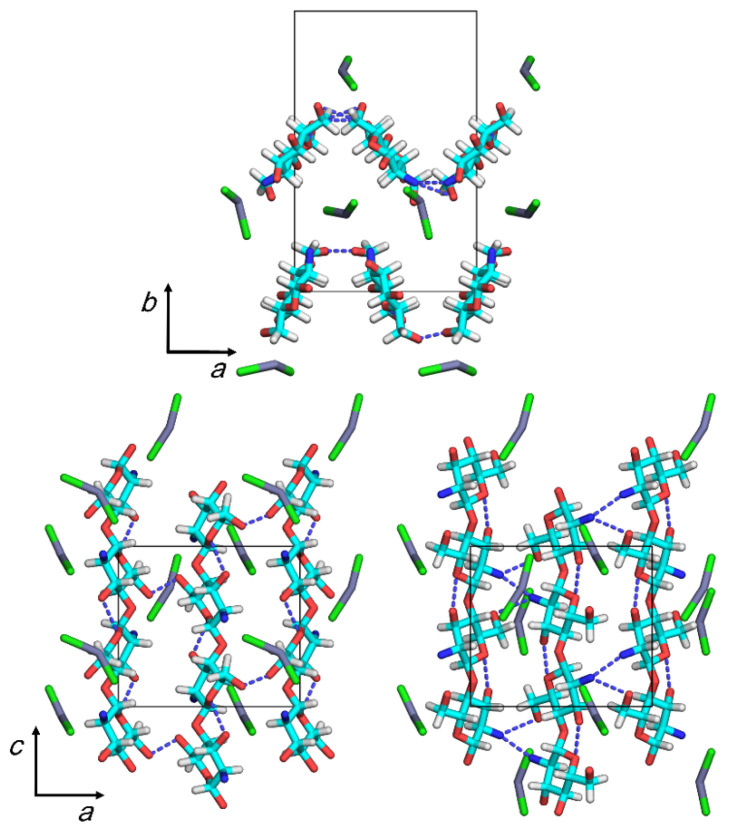
Projections of the proposed crystal structure on the *ab* (top) and *ac* (bottom) base planes (left: the chain sheet consisting of chains 1 and 2; right: the chain sheet consisting of chains 3 and 4). Dashed lines indicate hydrogen bonds.

**Figure 5 nanomaterials-11-01407-f005:**
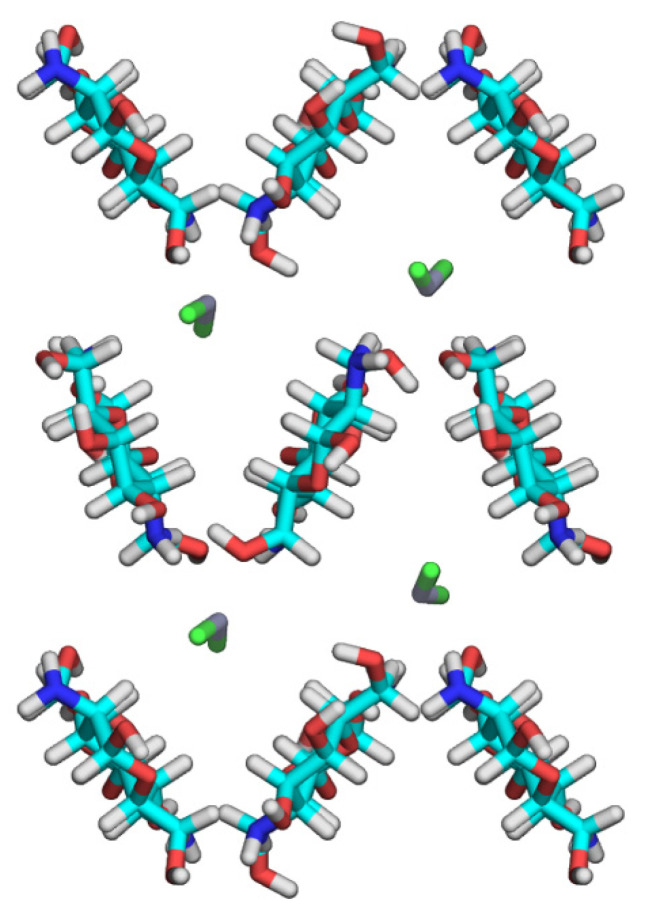
Projections of the SEQM-optimized structure of the crystal model on the *ab* base planes.

**Table 1 nanomaterials-11-01407-t001:** Chain packing modes of chain packing models.

Model	Chain Positions		Chain Polarities ^1^	
	Chain 1 (*u*1, *v*1)	Chain 2 (*u*2, *v*2)	Chain 1	Chain 2
*P*2_1_/*a* space group				
**1**	0.0, 0.0	0.0, 0.5	up	up
**2**			up	down
**3**	0.25, 0.25	0.25, 0.75	up	up
**4**			up	down
*P*2_1_/*b* space group				
**5**	0.0, 0.0	0.5, 0.0	up	up
**6**			up	down
**7**	0.25, 0.25	0.25, 0.75	up	up
**8**			up	down

^1^ When the position of C1 is higher than that of C4 along the fiber axis, a chain polarity is up. Otherwise, a chain polarity is down.

**Table 2 nanomaterials-11-01407-t002:** Summary of *ab* projection structure analysis.

Hydroxymethyl Conformation	*R*_w_ (*d*_N2–Zn_ in nm)			
μ1-μ2 refinement				
*gg*	0.310 (0.22)			
*gt*	0.289 (0.22)			
*tg*	0.290 (0.22)			
*d*_N2–Zn_ search				
*gg*	0.192 (0.23)	0.392 (0.36)	0.302 (0.45)	
*gt*	0.185 (0.25)	0.150 (0.30)	0.199 (0.37)	0.208 (0.45)
*tg*	0.214 (0.23)	0.196 (0.40)	0.202 (0.46)	

**Table 3 nanomaterials-11-01407-t003:** Chain packing parameters and *R*_w_ values of models selected in three-dimensional structure search.

Model	*R* _w_	*d* _N2–Zn_	Chain Packing Parameters			
			μ1 (deg.)	*w*1 (frac.)	μ2 (deg.)	*w*2 (frac.)
1	0.259	0.30	−11.7	0.068	116	−0.108
2	0.258	0.37	−2.91	−0.053	109	−0.174
3	0.245	0.46	−4.78	−0.214	109	−0.138

**Table 4 nanomaterials-11-01407-t004:** Chain packing parameters, *R*_w_, and nonbonding contacts of final refined models.

Model	ZnCl_2_ Linking Pattern	Contact σ	*R* _w_	*d*_N2–Zn_ (nm)	Chain Packing Parameters			
					μ1 (deg.)	*w*1 (frac.)	μ2 (deg.)	*w*2 (frac.)
1	1	83	0.246	0.34	3.61	−0.119	102	−0.243
2	5	138	0.244	0.40	−4.07	−0.116	106	−0.215
3	6	147	0.247	0.40	−0.402	−0.063	102	−0.209
4	1	52	0.245	0.45	12.1	−0.388	103	−0.345
5	2	172	0.245	0.46	11.5	−0.230	102	−0.169
6	5	193	0.244	0.46	−2.84	−0.214	107	−0.168
7	6	151	0.247	0.47	−2.02	−0.215	111	−0.154

**Table 5 nanomaterials-11-01407-t005:** Representative parameters of the final structure.

**Hydroxymethyl Conformations, χ_O6_**				
**Labels**		**Chain No.**		**Values (deg.)**
χ_O6A_		1, 2		171
χ_O6B_		1, 2		170
χ_O6A_		3, 4		171
χ_O6B_		3, 4		84.8
**Chain Packing Parameters, μ and *w***				
**Labels**		**Chain no.**		**Values**
μ1/deg.		1, 3		14.01
*w*1/frac.		1, 3		−0.3853
μ2/deg.		2, 4		102.1
*w*2/frac.		2, 4		−0.3431
**Distances of Intramolecular Hydrogen Bond**				
**Atom Labels**				**Values (nm)**
O3a/b		O5a/b		0.272
**Distances of Intermolecular Hydrogen Bonds**				
**Atom Labels**	**Chain No.**	**Atom Labels**	**Chain No.**	**Values (nm)**
O6a	1	O6b	2	0.198
O6b	1	O6a	2	0.220
O6a	3	N2a	4	0.285
N2b	3	N2a	4	0.256
O6b	3	N2b	4	0.286
N2a	3	O6a	4	0.285
N2a	3	N2b	4	0.256
*R* _w_				0.247
contact σ				39

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
