# Peer review of "Molecular and Crystal Structure of a Chitosan−Zinc Chloride Complex"

_nanomaterials, 2021, doi:10.3390/nano11061407_

Round 1

Reviewer 1 Report

This is an interesting topic. I appreciate the amount of work that was put into producing this manuscript. Well done!

Author Response

Thank you for your favorable comment which is encouraging.

Reviewer 2 Report

Manuscript presents high originality and novelity. In my opinion is prepared accordingly to rules of scientific excellence. 

Minor comments:
 1) One commonly finds chitin in the exoskeleton or cuticles of many invertebrates, and the cell walls of most fungi and some algae. -  please note that chitin has been confirmed in marine sponges 10.1016/j.msec.2019.110566 

2) why did you not use commercial chitosan with known DA degree and MW? please indicate DA degree and MW of chitosan obtained in your study.

Author Response

Thank you for your advices.

1) We briefly mentioned the chitin resources from marine sponges in the lines 27-29 with the reference recommended.

2) The sample used for X-ray measurement was characterized in the previous work reported in ref 20. We added the DA and Mw according to ref.20 in the lines 89-91 of the revised manuscript.

The reason why we did not use the commercial chitosan sample was that tendon chitin corresponds to the well-oriented sample, so that in-situ solid-state deacetylation of it readily gave an oriented chitosan sample which was much suitable to X-ray measurement than using the commercial chitosan sample̶—you must go through a film preparation and its stretching.

Reviewer 3 Report

This manuscript reported a study on the molecular structure of chitosan–ZnCl2 complex using x-ray diffraction. X-ray diffraction is a very common technique to study microstructure of crystals and chitosan-ZnCl2 is not a novel material. Moreover, I cannot find any new techniques in analyzing the experimental results. The main weaknesses of this work are that there is no information about the application of the chitosan–ZnCl2 complex in industry and science; and there is no information about how the experimental results of this microstructural structure can benefit the field/application. To sum up, this work is not suitable published in Nanomaterials but a journal regarding microstructure in chemistry.

Author Response

We appreciate your comment.

Our manuscript was intended to being submitted to the special issue "Emerging Functions of Nano-Organized Polysaccharides" and its aim should suite the topic of “crystalline-structure-triggered novel functions of nano-organized polysaccharides”.

Since the scope of the journal concerns the reviewer, it is up to the editor’s decision.